# Concentrations, Possible Sources and Health Risk of Heavy Metals in Multi-Media Environment of the Songhua River, China

**DOI:** 10.3390/ijerph17051766

**Published:** 2020-03-09

**Authors:** Kunyang Li, Song Cui, Fuxiang Zhang, Rupert Hough, Qiang Fu, Zulin Zhang, Shang Gao, Lihui An

**Affiliations:** 1International Joint Research Center for Persistent Toxic Substances (IJRC-PTS), School of Water Conservancy and Civil Engineering, Northeast Agricultural University, Harbin 150030, China; kunyleee@163.com (K.L.); ZhangFuxiang823@163.com (F.Z.); ijrc_pts_neau_paper@yahoo.com (Q.F.); 13204665308@163.com (S.G.); 2Research Center for Eco-Environment Protection of Songhua River Basin, Northeast Agricultural University, Harbin 150030, China; 3The James Hutton Institute, Craigiebuckler, Aberdeen AB15 8QH, UK; Rupert.Hough@hutton.ac.uk (R.H.); Zulin.Zhang@hutton.ac.uk (Z.Z.); 4State Key Laboratory of Environmental Criteria and Risk Assessment, Chinese Research Academy of Environmental Sciences, Beijing 100012, China

**Keywords:** heavy metals, Songhua River, multi-media environment, health risk, source apportionment

## Abstract

Heavy metal pollution in the river environment has been a source of widespread interest due to potential threats to human health and ecosystem security. Many studies have looked at heavy metal pollution in the context of single source-pathway-receptor relationships, however few have sought to understand pollution from a more wholistic multi-media perspective. To investigate potential risks in a more wholistic way, concentrations of six heavy metals (Cd, Cr, Ni, Cu, Zn and Pb) were detected in multi-media (water, sediment and riparian soil) collected from 14 sampling sites in the main stream of the Songhua River. Chemical analyses indicated that the average concentration of heavy metals in water followed: Zn > Cr > Cu > Pb > Ni > Cd, with a different trend observed in sediments and riparian soil: Zn > Cr > Ni > Pb > Cu > Cd. The potential risk was evaluated using the heavy metal pollution index (*HPI*), Nemerow pollution index (*P_N_*), hazard index (*HI*) and carcinogenic risk (*CR*) metrics. Results showed that all *HPI* values were lower than the critical level of 100 indicating that the levels of these targeted heavy metals were within drinking water safety limits. The *P_N_* indicated that both sediment (2.64) and soil (2.95) could be considered “moderately polluted”, with Cd and Zn providing the most significant contributions. A human health risk assessment suggested that the non-carcinogenic risks were within acceptable levels (*HI* < 1), as was the cancer risk associated with dermal adsorption (*CR* <10^−6^). However, the *CR* associated with ingestion exposure (4.58 × 10^−6^) exceeded the cancer risk threshold (10^−6^) indicative of elevated cancer incidence in exposed populations. Health-risk estimates were primarily associated with Cd in the Songhua River. Source apportionment was informed by Pearson correlation analysis coupled with principal component analysis (*PCA*) which indicated that Cu was mainly derived from natural (geogenic) sources; Cr and Ni were associated with industrial emissions; Pb might be derived from agricultural and transportation sources; Zn might be from industrial, agricultural activities and transportation; while Cd is likely from industrial and agricultural emissions. The source apportionment information could provide the basis for a risk-management strategy focused on reducing Cd and Zn emissions to the riverine environment. Results from this study will provide the scientific knowledge that is needed for measuring and controlling heavy metals sources and pollution characteristics, and identifying the potential cancer risk with different exposure pathways, as well as making effective environmental management policies at catchment or regional scales.

## 1. Introduction

Heavy metals are ubiquitous pollutants in the environment, derived from both geogenic and anthropogenic sources [1,2,3]. Heavy metals have been the subject of significant attention due to their environmental toxicity, bioaccumulation and persistence [2,3]. Excessive emissions and accumulation of heavy metals can cause serious pollution to environmental media such as air, soil, water and sediments, with potential implications for ecological safety and human health [4,5,6].

Rivers are an extremely important freshwater resource and ecological system that humans are often reliant upon [1,6,7]. However, rapid development of industry, agriculture and urbanization often results in polluting discharges (e.g., wastewater, agricultural run-off) to watercourses that impact water quality, with knock-on implications for ecosystem and human health [3,8,9]. Sediments are generally considered as both “source” and “sink” for most heavy metals and other pollutants in the water environment, thus measurements made in sediments can be a useful indicator of potential ecological risk to the aquatic environment [10,11,12,13]. Similarly, soil can also be an important storage reservoir of pollutants, which can enter soil through wastewater discharge, atmospheric deposition and fertilizer application [14,15,16]. Contaminated soils can subsequently pollute the riverine environment via runoff to surface water and via leaching to groundwater [16,17]. Therefore, water, sediment, and soil are not independent media, but components of a wholistic system that people are also a component of. People depend on this environment for basic provisioning services and are, therefore, simultaneously impacted through exposure to any heavy metals present in that environment.

The human health risk assessment approach developed by the United States Environmental Protection Agency (USEPA) has been used extensively to estimate the potential health risk [4,5,6,18]. In general terms, this is a hazard-based approach which defines ‘risk’ as the ratio between an exposure and a pre-determined ‘safe’ level for that exposure. These ‘safe’ levels are sometimes outcome-specific cf. reference dose vs. cancer slope factor. In China, the USEPA methodology has been used to estimate human health risks associated with specific high-profile water pollution events in the Yellow River, Pearl River, Yangtze River and Xiangjiang River [4,7,9,17]. However, these studies are limited by adopting the single source-pathway-receptor paradigm of the USEPA methodology which can poorly reflect the comprehensive human health risk [4,5,14]. In non-occupational settings, it has been demonstrated that exposure is not dominated by a single pathway or exposure media [19]. Therefore, a comprehensive human health risk assessment that integrates all important exposure pathways within the multi-media environment is needed to improve confidence in risk estimates.

The Songhua River is one of the seven largest rivers in China, and an important source of drinking and irrigation water, as well as water abstraction for industrial production [19,20]. The land use of the Songhua catchment is dominated by intensive cereal production and also includes a number of primary and secondary heavy industries (petrochemical, coal mining, mechanical processing, etc.) [20,21,22,23]. Discharges and run-off from these industries to both surface water and groundwater often contain heavy metals. Long-term exposure can have adverse impacts on the environment and human health. Research on heavy metals in the Songhua River Basin has received much attention, but is currently restricted to studying the ecological risks and impacts associated with heavy metals in sediments [21,22]. Therefore, there is an imperative to widen our knowledge to include multi-media (water, sediments and soils) sources and sinks of heavy metals and to understand potential exposure and risk levels from a more comprehensive range of pathways. Hence, the purpose of this study is: (1) to analyze the characteristics of heavy metal pollution in water, sediments and riparian soil of the Songhua River; (2) to assess the overall pollution level of heavy metals in using the heavy metal pollution index (*HPI*) method and the Nemerow pollution index method (*P_N_*); (3) to estimate daily heavy metal intake (*CDI*) under different exposure pathways, and evaluate associated health risks for residents of the riparian zone; and (4) to analyze possible sources of pollution using Pearson correlation analysis coupled with principal component analysis (*PCA*). The results of this study will provide the scientific basis for developing a multi-media, multi-pathway exposure risk model. For the specific case of heavy metals, it provides a scientific rationale for identifying their spatial distribution and sources within the multi-media environment, as well as the important exposure pathways and associated risks.

## 2. Materials and Methods

### 2.1. Study Area

The Songhua River, the largest tributary of Heilongjiang River in China, flows through Heilongjiang Province and Jilin Province [20]. The average annual temperature is 3–5 °C, and the freezing period is 5 months (from November to March next year) in the watershed [21]. The main stream of Songhua River (124°39′–132°31′ E and 45°26′–47°43′ N) has a drainage area of approximately 1.893 × 10^5^ km^2^ and a total length of 939 km. The abundant incoming water is the main irrigation water source in the Sanjiang Plain and the Songnen Plain [19,20]. At the same time, the Songhua River Basin is an important catchment for agricultural production (primarily cereals and other combinable crops), as well as being an industrial and energy base in China [21,23]. Influenced by the topographical features of the main stream of the Songhua River and the distribution of major cities and counties along the river bank, 14 sampling points were selected in July 2015, as shown in Figure 1 and Appendix A (presented in Appendix A).

### 2.2. Heavy Metal Analysis

#### 2.2.1. Sample Collection and Processing

We refer to the Technical Specifications for Surface Water and Wastewater Monitoring (HJ_T91-2002) [24]. Five sub-sampling points within 30 m of each pre-set sampling location (Figure 1) were randomly selected. From each location, water was collected using polyethylene plastic bottles previously washed with nitric acid. During sampling, care was taken to avoid agitation of sediment on the riverbed. Collected samples were acidified with Nitric acid to bring the pH down to less than 2. Surface sediment (0–10 cm) was collected using a grab sampler from the 14 sampling sites using the same approach as for the water samples. For sediments, the five sub-samples were combined into a single representative sample per site. Each sample was stored in a pre-washed glass container with a Teflon cap. At the same time, riparian soil samples (0–20 cm) were also collected within 30 m at water sampling sites using a stainless steel scoop that had been prewashed with deionized water and stored in a pre-washed glass container. Any debris such as weeds and gravel from solid samples (sediment samples were allowed to stand before pouring off the overlying water) was removed manually. All samples were transported to the International Joint Research Center for Persistent Toxic Substances laboratory at Northeast Agricultural University (IJRC-PTS, NEAU) in Harbin where they were stored in a refrigerator prior to analyses.

Sample processing was undertaken in accordance with the Environmental Quality Standard for Soil Environmental Quality Risk Control Standard for Soil Contamination of Agricultural Land (GB15618-2018) [25] and Water and Wastewater Monitoring and Analysis Methods (Fourth Edition) [26]. Briefly, soil and sediment samples were air-dried at room temperature and passed through a 20 mesh (0.84 mm) nylon screen. Sieved samples were then ground to a particle size of less than 0.25 mm with an agate mortar. 500 mL water samples were placed on a heating plate and concentrated by evaporation to 50 mL for use. Soil and sediment samples were digested in a Teflon crucible by the wet oxidation method (HCl-HNO_3_-HClO_4_-HF) (GR, Tianjin Yaohua Chemical Reagent Co., Ltd., Tianjin, China) until no obvious solid particles remained [8]. Digestates were assayed for heavy metal content using a Thermo Fisher Scientific atomic absorption spectrophotometer (ICE 3500), the flame portion was used to determine the contents of Cu, Cr, Zn, and Ni, and the graphite furnace portion was used to determine the contents of Cd and Pb.

#### 2.2.2. Quality Assurance and Quality Control (QA/QC)

Strict quality assurance and control of all analytical data was conducted. The glass and polyethylene vessels used were fully soaked in 2 mol L^−1^ HNO_3_ for more than 24 h, washed with ultrapure water and dried before use. The water used in the analysis was ultrapure water and the reagents used were excellent grade pure. Blank and standard samples were digested and analyzed using the same procedure. Standard reference materials (GBW07305 and GBW07458) were obtained from the Chinese Academy of Measurement Science. The recovery rate ranged from 93.7% to 102.5%. The standard deviations between parallel samples were less than 5%. It was guaranteed that the correlation coefficient of calibration curves of the 6 heavy metals was greater than 0.9995.

### 2.3. Evaluation Pollution of Heavy Metals

#### 2.3.1. Heavy Metal Pollution Index

The heavy metal pollution index (*HPI*) indicated the relative total water quality determined based on all selected parameters [27]. The *HPI* provides an estimate of the comprehensive impact of each individual heavy metal on the overall water quality, and is determined as follows:
(1)$$HPI=\frac{\sum \left(Q_{i}W_{i}\right)}{\sum W_{i}}$$
(2)$$Q_{i}=\frac{C_{i}}{S_{i}}\times 100;\;W_{i}=\frac{k}{S_{i}}$$
where *Q_i_* is a sub-indicator of the heavy metal pollution index, *W_i_* is the unit weight of the heavy metal parameter, *C_i_* is the concentration value of the *i*-th heavy metal parameter (mg L^−1^), and *S_i_* is the highest standard permissible value of the *i*-th parameter, The Standards for Drinking Water Quality (GB5749-2006) [28] was selected as the source of the highest allowable level for each of the heavy metal of interest. The proportional constant (*k*) was set to 1 [3]. The allowable and critical *HPI* value of drinking water is usually 100 [27]. Based on this, the categories of *HPI* are summarized in Appendix A.

#### 2.3.2. Nemerow Pollution Index

The Nemerow pollution index (*P_N_*) can not only comprehensively reflect the pollution level of a given area, but also highlight the environmental hazard of the most significant pollutants within a given sampling scheme [29,30,31]. Values of *P_N_* for soil and sediment can be calculated using the following equations:
(3)$$P_{N}=\sqrt{\frac{{P^{2}}_{i(\max)}+P_{i(ave)}{}^{2}}{2}}$$
(4)$$P_{i}=\frac{C_{i}}{S_{i}}$$
(5)$$P_{i(ave)}=\frac{1}{n}\sum_{n}^{1}P_{i}$$
where *P_N_* is the pollution index of each sampling site, *P_i_* is the single contamination factor of the *i*-th heavy metal, *P_i(ave)_* is the arithmetic mean of the single contamination factor of all heavy metals, *P*_*i*(*max*)_ the maximum contamination factor among the heavy metals, *C_i_* is the measured concentration of heavy metal and *S_i_* represents the quality standard value of different heavy metals. The risk screening values from the background value of soil in Heilongjiang province [32] were used for *S_i_*. Values of *S_i_* for Cu, Cr, Zn, Pb, Ni, and Cd were thus 20, 58.6, 70.7, 24.2, 22.8 and 0.086 mg kg^−1^, respectively. The evaluated criteria of *P_N_* are classified in Appendix A.

#### 2.3.3. Human Health Risk Assessment

Ingestion and dermal absorption, the most common and important exposure pathways for soil and water in the living environment [3,14], were selected for human health risk assessment. The US Environmental Protection Agency (USEPA) points out that the amount of pollutants absorbed by the human body is calculated based on chronic daily intake (*CDI*) [33].

For the water and soil, the *CDI* was defined as Equations (6)–(9) [5,33]:
(6)$$CDI_{w\text{-}in}=\frac{C_{i}\times IR\times ABS_{g}\times EF\times ED}{AT\times BW}$$
(7)$$CDI_{w\text{-}derm}=\frac{C_{i}\times SA\times K_{p}\times ABS_{d}\times ET\times EF\times ED\times CF}{AT\times BW}$$
(8)$$CDI_{s\text{-}in}=\frac{C_{i}\times IR\times EF\times ED}{AT\times BW}\times CF$$
(9)$$CDI_{s\text{-}derm}=\frac{C_{i}\times SA\times SL\times ABS_{d}\times EF\times ED}{AT\times BW}\times CF$$
where *CDI* refers to the exposure doses from ingestion and dermal absorption (mg kg^−1^d^−1^), *C_i_* was the average concentration of the heavy metal *i* in water (mg L^−1^), *CF* was conversion factor. The remaining parameters and source of parameters are shown in Table 1.

The cancer risk (*CR*) indicates the incidence of cancer that exceeds expected levels in a person’s lifetime by exposure to some certain carcinogenic substances with Equation (10):
(10)CR=CDI×SF
where *SF* is the cancer slope factor (mg^−1^ kg day). The acceptable level of a *CR* value was usually between10^−6^ and 10^−4^ [35]. When values of *CR* exceed 10^−4^ this indicates a high cancer risk to humans, while values of *CR* between 10^−6^ and 10^−4^ indicate a lower but elevated cancer risk.

The potential non-carcinogenic risks were assessed by hazard quotient (*HQ*) as Equation (5) [35]. When values of *HQ* were >1, non-carcinogenic effects should be considered.
(11)$$HQ=\frac{CDI}{RfD}$$
where *RfD* represents the heavy metal intake reference dose (mg kg^−1^ day^−1^), in this study all values of *RfD* used were those published by the USEPA [36]. It should be noted that all *RfDs* published by the USEPA use dose estimates based on typical American body weights and are, therefore, likely to underestimate risk in Asian populations [18]. In addition, the total potential non-carcinogenic risks caused by different pathways were assessed by Hazard Index (*HI*) as Equation (12):
(12)$$HI=\sum HQ=HQ_{derm}+HQ_{in}$$
where *HQ_derm_* and *HQ_in_* represent the hazard quotient (*HQ*) caused by dermal absorption and ingestion pathway, respectively. Similarly, if the *HI* were >1, adverse effects on human health should be considered [35]. Both *HI* and *CR* were calculated using the parameters summarized in Table 1 to reflect the potential health risks of the residents in the study area.

#### 2.3.4. Statistical Analysis

The normality test was performed by Kolmogorov–Smirnov test for each trace metal; One-sample T-test was used to compare the concentrations of heavy metals in individual environmental media with background value; The inter group difference was compared with the independent sample T-test; Pearson correlation analysis was combined with principal component analysis (*PCA*) to identify the source of pollution. Prior to this, the validity of *PCA* was tested by Kaiser–Meyer–Olkin (*KMO*) values (>0.5) and Bartlett Sphericity test (*p* < 0.01) [12,19], All mathematical and statistical calculations were performed by Excel 2016, Origin 8.6 and SPSS 21.0, with sample distributions plotted using ArcGIS 10.2.

## 3. Results and Discussion

### 3.1. Concentrations and Spatial Distribution of Heavy Metals

#### 3.1.1. Concentration

The heavy metal contents and statistical characteristics of the Songhua River water, sediments and riparian soil are shown in Table 2. The average concentrations in water followed: Zn > Cr > Cu > Pb > Ni > Cd, and these levels were significantly higher than the background concentrations found in the Songhua River (*p* < 0.01) [37]. For example, Zn (64.25 μg L^−1^) and Cr (12.10 μg L^−1^) were 16.56 and 14.13 times as high as the background values [37]. The concentration of Cd was at least 0.26 μg L^−1^ and 4.07 times greater than the reported background value. Levels of Zn in the Songhua River water were even higher than those reported from the Hun River [38], a catchment strongly impacted by sewage sludge applications to land, but were generally lower than levels of Zn measured in the Xiangjiang River flowing through Hunan Province (in which there are non-ferrous metals) [4]. All monitoring activities reported on here were in compliance with the requirements of the Environmental quality standards for surface water (GB3838-2002) for the secondary protection zone of centralized drinking water for surface water source [39]. Although there are elevated levels of pollution in the Songhua River compared to the reported background values, water quality is still adequate for irrigation based on current standards.

The average concentrations of heavy metals in sediments were: Zn > Cr > Ni > Pb > Cu > Cd. The contents of heavy metals in sediments such as Zn (175.76 mg kg^−1^) and Cd (0.28 mg kg^−1^) were significantly higher than the background concentrations in the soils of Heilongjiang Province (*p* < 0.01) [32]. A number of monitoring sections returned concentrations of Cd (S1, S9, S10, S12) and Zn (S1, S7, S10, S12) in excess of the risk screening value of the Environmental Quality Standard (GB15618-2018) [25]. Concentrations of Pb were lower than those reported in a previous study of the Songhua River, while levels of Zn were greater [40]. Compared to the water analysis, the relative concentrations of the heavy metals in sediment still indicated that Zn and Cd were at the extremes of the distribution; however, some differences were seen in the relative ranking of the other 4 metals. Heavy metals in sediments tend to be the result of long-term accumulation [12,13], while heavy metal contents in surface water more closely reflect contemporary pollution within the catchment, thus explaining the relative differences seen between water and sediments. Compared with sediments in other rivers (Appendix A), the contents of Cu, Cr, Zn, Ni and Pb in the Songhua River were significantly lower than those in the Yangtze River (*p* < 0.01) [11], while the concentrations of heavy metals were higher than those in many major European rivers such as the Ebro River and Seine River [41,42].

The average concentration of heavy metal in the riparian soil was Zn > Cr > Ni > Pb > Cu > Cd, and consistent with the trends seen in the sediment data. The contents of Ni, Pb and Cu were close to background values reported for Heilongjiang Province [32], however, Cd (0.31 mg kg^−1^) and Zn (145.83 mg kg^−1^) were 3.59 and 2.06 times higher than their background values, respectively (*p* < 0.01). These monitoring data suggest that both Cd and Zn have accumulated to a certain extent. The average concentration of Cd measured in this study was higher than the median value, indicating that there was a sampling location or locations with significantly elevated levels of Cd pollution, such as S6 and S10, which had significant leverage on the dataset. The coefficient of variation of Cd in soil was higher than that of sediment and water, which may suggest some differences in sources. With the exception of Zn, the contents of heavy metals in soil were higher than those in sediments in the Songhua River, and this might be associated with atmospheric deposition. A study on heavy metal sources in the Songnen Plain of Heilongjiang Province found that atmospheric deposition was a significant input pathway by which Cd, Cu, Pb and Zn enter surface waters, accounting for 78%–98% of the total input [43].

#### 3.1.2. Spatial Distribution

The spatial distribution of heavy metal contents of water, sediments and riparian soil in the Songhua River is shown in Figure 2. The maximum concentration of Zn and Cd were found in water at Harbin section (S4 and S5), which was significantly higher than other locations (*p* < 0.01), while the highest concentrations of the other heavy metals were detected in Yilan County (S9) characterized by agricultural non-point sources as well as a few point source discharges. For the sediment at site S9, the concentrations of Cd, Cu and Zn are evaluated as being ‘high level’. Given these metals are used as an additive in commonly used insecticides and fertilizers, the inference is that agricultural activities are likely important pollution sources in this location. Considering the sediment samples, points S3 and S8 are located in rural areas where the heavy metal contents were far lower compared to samples from S4, S5, S11 and S12 that are all close to urban areas (*p* < 0.05). The proximity of urban areas to the riparian zone influences both the magnitude and spatial distribution of heavy metal pollution [44].

### 3.2. Pollution Assessment of Heavy Metals

The Songhua River is a major source of both potable and irrigation water for residents within the riparian zone. Therefore, the most stringent drinking water guidelines (GB5749-2006) and background values were taken as references in this study [28,32]. The distribution of the *HPI* and *P_N_* are shown in Figure 3. The *HPI* of water samples from the Songhua River was less than 100, indicating that the water is suitable for drinking. [27]. Given that Pb and Cd can cause both acute and chronic toxic effects on the human body even at very low levels, and thus they were considered as the main contributors to the *HPI* in the present study, with average contribution rates of 60.4% and 21.6%, respectively. The average *HPI* was 13.5, which was at the “low” level, but it was close to the “medium” level (*HPI* = 15). Downstream of Harbin City, levels of heavy metal pollution gradually increased (S3–S6), indicating that intensive human activities could lead to the increasing trace elements in the water. The *HPI* value was especially acute at location S6 which was influenced by Harbin City (upstream) as well as by diffuse sources from adjacent agricultural land.

According to the pollution assessment standard proposed by Nemerow [29], the average of the single factor pollution index (*P_i_*) of the six heavy metals in sediments and riparian soil has the same order: Cd > Zn > Cr > Ni > Cu > Pb (Appendix A), indicating “Severe pollution” by Cd (*P_i_* > 3) and “moderate pollution” by Zn (2 < *P_i_* ≤ 3). Overall, levels of pollution in the riparian soils were greater than those derived for the sediment samples. The *P_i_* value of Cd was significantly elevated compared to the other heavy metals (*p* < 0.05). Cd is generally derived from industrial emissions and coal combustion, as well as the application of fertilizers (especially for phosphate fertilizers) [4,8]. The *P_N_* values are an indicator of the comprehensive pollution of multiple heavy metals. The *P_N_* of soil and sediments for all sampling sites ranged from 1.41 to 7.46 and 1.39 to 5.07, with an average of 2.95 and 2.64, respectively, which were considered to be “moderately polluted” (2.5 < *P_N_* ≤ 7). The results indicated that there were locations where pollution levels could be considered to be significantly elevated including riparian soils from S6, S8 and S10 (*p* < 0.05) and sediments from S1, S9, S10 and S12 (*p* < 0.05). It might be strategic to focus on these locations as part of future environmental monitoring and pollution prevention efforts. In addition, the *P_i_* of Cd in riparian soil is 3.54 times greater than the *P_i(ave)_* at site S10, which suggests the presence of a specific point source or sources. Overall, values of *HPI* of water indicated that Cd and Pb were the main contributors to the derived level of pollution; while values of *P_N_* indicated that Cd and Zn were the most important pollution factors both in sediments and riparian soil.

### 3.3. Human Health Risk Assessment

Among the environmental media we studied, surface water and soil were directly exposed to pollutant inputs and also acted as secondary sources, as well as being important media as carriers of pollutants within specific human exposure pathways. Indices of health risk including *HI* and *CR* were calculated for ingestion and dermal absorption exposure pathways (Table 3). Despite Cr and Pb accounting for the bulk of the non-carcinogenic risk, values of *HI* were considered to be within acceptable levels. Although the *HPI* and *P_N_* indicated that levels of Pb pollution in the catchment were low, its potential health risk to humans was indicated as being more significant highlighting the non-linear relationship between environmental levels and the magnitude of exposure. Given the importance of Cd as the main driver of the carcinogenic risk estimates, the *CR* values of Cd were only calculated for cancer risk assessment in this study. Although the cancer risk from dermal absorption (1.08 × 10^−7^) could be considered to be within appropriate limits of safety, the cancer risk from ingestion (average of 4.58 × 10^−6^) across all sampling sites indicates elevated cancer risk. Values of *CR* for ingestion exceeded 10^−6^ at every single sampling location.

Compared with dermal absorption, ingestion was the primary human exposure pathway for heavy metals in the Songhua River Basin, with exposure from water ingestion being slightly larger than that from the riparian soil. Risks associated with cancer and non-carcinogenic outcomes from the Songhua River were overall lower than those reported for the Yangtze River and Huaihe River, but higher than those of the Liujiang River [45,46,47]. The *CR* of heavy metals in water (*CR_w-in_* = 1.60 × 10^−6^) indicated a possible elevated cancer risk associated with consuming water from the Songhua River, especially the case for water abstracted from the Harbin section that had the highest levels of Cd reported in this study. While direct ingestion is likely to be the most significant exposure route, this study cannot be viewed as an overall health risk assessment due to other pathways such as dietary and inhalation exposure being omitted at present.

### 3.4. Analysis of Sources of Heavy Metal Pollution

Identifying sources of heavy metals is critical to effectively reducing pollution and human health risks [1,5]. In this study, Pearson correlation analysis coupled with principal component analysis (*PCA*) was used to analyze the sources of heavy metals in the water and sediments of the Songhua River. The variables were further explained by varimax rotation.

Three factors were extracted from the water, accounting for 80.6% of the total variance. As illustrated in Figure 4a. Factor 1 (PC1), which had the highest cumulative contribution rate accounting for 49.9% of the variance, was heavily weighted by Cu, Cr, and Ni (loadings were greater than 0.80). There were significant correlations among these three elements (Appendix A), while Cu, Cr, and Ni were significantly higher than the environmental background values (*p* < 0.01) [37]. Industrial production activities such as metalworking, electroplating and machinery manufacture use raw materials containing heavy metals such as Cr, Cu and Ni [8,31,44]. Therefore, it is inferred that PC1 represented the influence of the industrial production activities. Factor 2 (PC2), had a high loading of Pb (0.81) and Zn (0.68) and accounted for 16.9% of the total variance. Albasel and Cottenie had found that the contents of Pb and Zn rapidly decrease with increasing distance from roads [48]. Automobile exhaust and the wear of vehicle components will cause the accumulation of heavy metals, especially the content of Pb which until recent times was closely related to motor vehicles [16]. In addition, the application of pesticides and insecticides containing Pb and Zn could affect environmental safety in water due to their input through soil runoff and return flows [8,49]. Therefore, it was suggested that PC2 might be mainly affected by traffic and agricultural sources. Factor 3 (PC3) had a high loading of Cd accounting for 13.8% of the total variance. Cd was generally associated with the industries including electronics, printing and dyeing, electroplating chemical industry, as well as from the excessive use of phosphate fertilizers [8]. Therefore, PC3 might be attributable to both industrial and agricultural sources.

For sediment, three factors were extracted, accounting for 82.9% of the total variance. Individually, Factor 1 (PC1) had the high loading of Cu, Ni, Cd and Cr accounted for 49.1% of the variance as illustrated in Figure 4b. The concentrations of Cu and Ni were lower than the background value at a majority of the sampling sites, indicating geogenic sources of Cu and Ni. Elevated concentrations of Ni were found within the industrial zones at S5 (Harbin), S9 and S12 (Jiamusi). Given that Ni, Cd and Cr were mainly used in electroplating, electronics, printing and dyeing [8,50], it was speculated that PC1 may represent a combination of natural sources and industrial emissions. The cumulative contribution rate of Factor 2 (PC 2) was 20.0%, with a high load of Pb, which may have largely originated from gasoline containing lead in the last century. Due to its environmental persistence, Pb is still an important indicator of transportation sources [21,48]. Moreover, the bioavailable form of Pb in the Songhua River sediments accounted for over 55%, and demonstrated the anthropogenic influences on Pb levels in the sediments [51]. Thus, the PC2 is likely from traffic sources. Factor 3 (PC3) had a high loading of Zn accounted for 13.8% of the variance. The concentration of Zn in the sediment samples is correlated to those measured in the water (*R* = 0.623, *p* < 0.05), indicating that they possibly have some similar sources (agriculture and transportation sources). In addition, Zn can be also derived from industrial processes such as mechanical processing, steel smelting, etc. [8,31]. It was concluded that the industrial, agriculture and transportation emissions together consist of the PC3.

The source apportionment confirmed that agriculture and industry were the main sources of heavy metals in the Songhua River basin. Thus, optimization and control of agricultural practices such as adopting precision agriculture approaches to chemical usage would aid in pollution mitigation. Due to the dominance of agricultural land use in the catchment, the potential for pollution reduction is considerable. At the same time, there are opportunities to adapt industrial processes and waste-management approaches to improve their environmental sustainability. This is especially pertinent to large urban areas such as Harbin. It is also imperative to improve the regulation of industrial units outside the main urban areas where clandestine discharges are likely to be more prevalent to improve the overall pollution level of locations such as S9.

## 4. Conclusions

The results of tests on the Songhua River water, sediments and soils confirmed that environmental pollution should be considered in a wholistic manner given the spatial variability of sources, interactions between pathways of exposure, as well as the non-linearity of the exposure term. Overall, the possibility of non-carcinogenic risk in the Songhua River was found to be very low. However, the cancer risk associated with consumption of the river water was slightly elevated above the cancer risk threshold. This cancer risk was mainly attributable to the presence of Cd in the water, and further work is required to understand the efficacy of the current water treatment regime for the removal or dilution of Cd and other potential pollutants associated with human health. The multi-media analysis indicated a significant accumulation of metals, particularly Cd and Zn, over time. Industrial emissions are likely to be the primary source of the observed heavy metal enrichment. Zn and Pb are also likely to have been derived from agricultural activities and transportation. Agricultural sources of Cd also cannot be ignored. Optimization and control of agricultural management with a focus on precision agriculture approaches could be one way to reduce pollution discharges. By adopting a more wholistic multi-media, multi-exposure approach to risk assessment, we have obtained a more comprehensive understanding of the interaction between local human activities and the riverine environment. This improved understanding helps aid mitigation responses as well as highlighting important knowledge gaps for future investigation.

## Figures and Tables

**Figure 1 ijerph-17-01766-f001:**
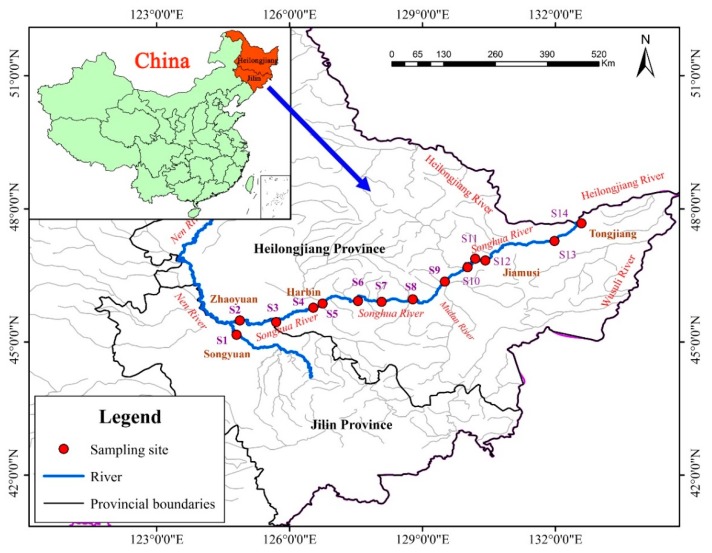
Locations of sampling sites of the Songhua River.

**Figure 2 ijerph-17-01766-f002:**
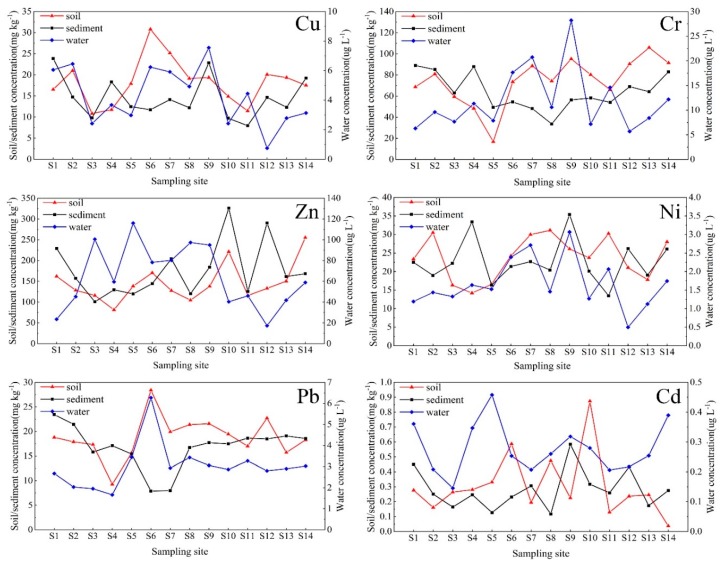
Concentration of Cu, Cr, Zn, Ni, Pb and Cd in the water, sediment and riparian soil.

**Figure 3 ijerph-17-01766-f003:**
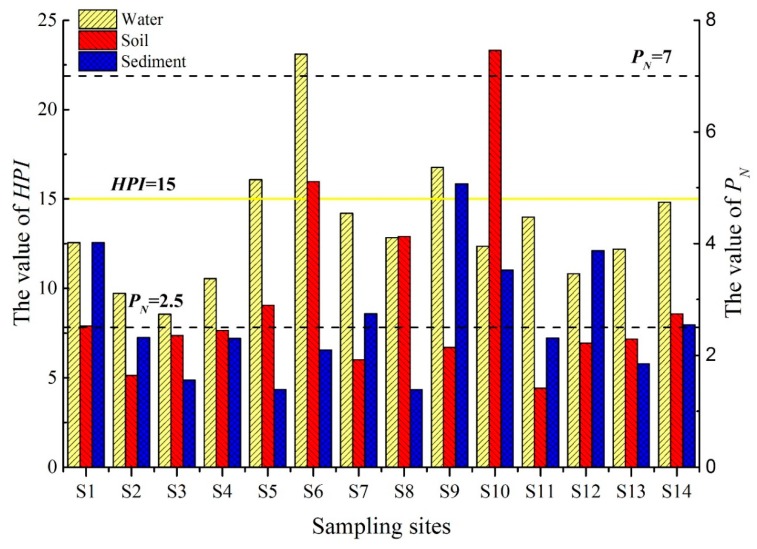
Heavy metal pollution index (*HPI*) of water, Nemerow pollution index (*P_N_*) of sediment and riparian soil in Songhua River.

**Figure 4 ijerph-17-01766-f004:**
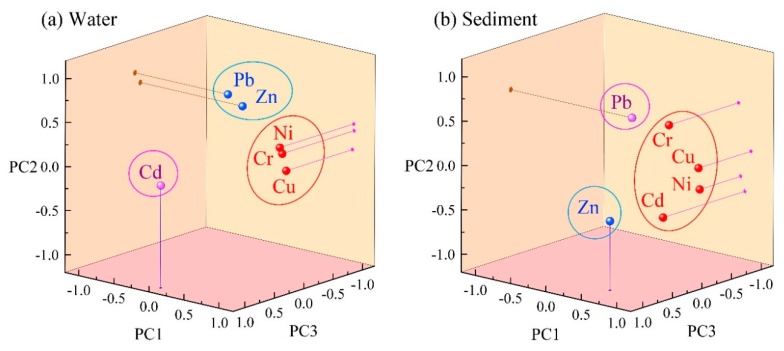
Load diagram of water (**a**) and sediment (**b**) principal component factor.

**Table 1 ijerph-17-01766-t001:** Exposure parameters in the health risk model.

Exposure Parameter	Water		Soil	
	Unit	Reference	Unit	Reference
Ingestion rate (*IR*) [19,34]	L d^−1^	1.227	mg d^−1^	100
Exposure frequency (*EF*)	day	365	day	365
Exposure duration (*ED*) [21]	a	74.8	a	74.8
Daily exposure time (*SL*) [35]	hr day^−1^	0.6	-	-
Average weight Body weight (*BW*) [19]	kg	63.1	kg	63.1
Average life time (*AT*) [21]	d	27,302	d	27,302
Conversion factor Conversion factor (*CF*)	L cm^−3^	10^−6^	mg kg^−1^	10^−6^
Skin exposed area Skin-surface area (*SA*) [35]	cm^2^	18,100	cm^2^	18,100
Permeability coefficient (*K_p_*) [35]	cm hr^−1^	Pb:10^−4^; Cd: 10^−3^; Cr: 2 × 10^−3^; Zn: 6 × 10^−4^; Ni: 2 × 10^−4^; Cu: 10^−3^	-	-
Gastrointestinal absorption factor (*ABS_g_*) [35]	-	Pb:0.117; Cd:0.05; Cu:0.3; Zn:0.2; Ni:0.2; Cr:0.038	-	-
Skin adhesion factor (*SL*) [35]	-	-	mg cm^−2^ d^−1^	0.2
Dermal absorption factor (*ABS_d_*) [35]	-	0.001	-	-

**Table 2 ijerph-17-01766-t002:** Statistical characteristics of heavy metal content in Songhua River.

Medium	Element	Range	Average	Median	Standard Deviation	Coefficient of Variation	Background Values [32,37]
Water μg L^−1^	Cu	0.75–7.55	4.27	4.06	1.90	44.5%	1.46
	Cr	5.71–28.23	12.01	10.10	6.16	51.3%	0.85
	Zn	17.29–116.01	64.25	59.13	29.49	45.9%	3.88
	Pb	1.66–6.27	3.02	2.92	1.04	34.4%	1.76
	Ni	0.50–3.07	1.68	1.49	0.66	39.2%	1.02
	Cd	ND–0.46	0.26	0.25	0.09	42.1%	0.06
Sediment mg kg^−1^	Cu	7.94–23.88	14.57	13.30	4.66	32.0%	20
	Cr	33.66–88.99	63.97	60.62	16.24	25.4%	58.6
	Zn	100.69–326.14	175.76	159.35	64.11	36.5%	70.7
	Pb	7.88–23.44	16.84	17.60	4.16	24.7%	24.2
	Ni	13.47–35.41	22.71	21.77	5.77	25.4%	22.8
	Cd	1.17–5.82	0.28	0.26	0.13	45.2%	0.086
Riparian soil mg kg^−1^	Cu	10.83–30.78	18.27	18.52	5.19	28.4%	20
	Cr	16.79–105.90	74.26	77.20	21.66	29.2%	58.6
	Zn	81.40–255.18	145.83	135.42	44.01	30.2%	70.7
	Pb	9.27–28.41	18.82	18.54	4.15	22.0%	24.2
	Ni	14.20–31.13	23.79	23.97	5.64	23.7%	22.8
	Cd	0.37–0.87	0.31	0.25	0.20	66.2%	0.086

**Table 3 ijerph-17-01766-t003:** Hazard quotient and carcinogenic risk for each element and exposure pathway.

Element	*RfD_in_*	*RfD_derm_*	*SF*	*CDI_w-in_*	*CDI_s-in_*	*CDI_w-derm_*	*CDI_s-derm_*	*CDI_in_*	*CDI_derm_*	*HQ_in_*	*HQ_derm_*	*HI*	*CR_in_*	*CR_derm_*
Cu	0.04	0.012	-	2.49 × 10^−5^	2.89 × 10^−5^	7.34 × 10^−10^	1.05 × 10^−6^	5.38 × 10^−5^	1.05 × 10^−6^	1.35 × 10^−3^	8.74 × 10^−5^	1.43 × 10^−3^	-	-
Cr	0.003	0.015	-	8.87 × 10^−6^	1.18 × 10^−4^	4.13× 10^−9^	4.26 × 10^−6^	1.27 × 10^−4^	4.26 × 10^−6^	4.22 × 10^−2^	2.84 × 10^−4^	4.25 × 10^−2^	-	-
Zn	0.3	0.06	-	5.00 × 10^−5^	2.31 × 10^−4^	6.64× 10^−9^	8.37 × 10^−6^	2.81 × 10^−4^	8.37 × 10^−6^	9.37 × 10^−4^	1.40 × 10^−4^	1.08 × 10^−3^	-	-
Pb	0.001	4 × 10^−4^	-	6.88 × 10^−6^	2.98 × 10^−5^	5.20 × 10^−11^	1.08 × 10^−6^	3.76 × 10^−5^	1.08 × 10^−6^	3.67 × 10^−2^	2.70 × 10^−3^	3.94 × 10^−2^	-	-
Ni	0.02	0.005	-	6.51 × 10^−6^	3.77 × 10^−5^	5.76 × 10^−11^	1.36 × 10^−6^	4.42 × 10^−5^	1.36 × 10^−6^	2.21 × 10^−3^	2.73 × 10^−4^	2.48 × 10^−3^	-	-
Cd	5 × 10^−4^	5 × 10^−6^	6.1	2.62 × 10^−7^	4.89 × 10^−7^	4.65 × 10^−11^	1.77 × 10^−8^	7.51 × 10^−7^	1.77 × 10^−8^	1.50 × 10^−3^	3.54 × 10^−3^	5.05 × 10^−3^	4.58 × 10^−6^	1.08 × 10^−7^

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
