# Peer review of "Concentrations, Possible Sources and Health Risk of Heavy Metals in Multi-Media Environment of the Songhua River, China"

_ijerph, 2020, doi:10.3390/ijerph17051766_

Round 1
Reviewer 1 Report
There are some comments which have to be considered before publication.
110: “(3) to estimate daily heavy metal absorption (CDI)…”
Comment: Better to remove CDI from the sentence or use the correct meaning. Chronic Daily Absorption was used for CDI (line 202); however, CDI is most commonly used for Chronic Daily Intake. Does the CDI mean Chronic Daily Absorption or Chronic Daily Intake?
130: Sample Collection and Processing
Comment: Give us more detail about the procedure and equipment used to collect the sediment.
187-189: Equations (3) and (5)
Comment: Are Pi(ave) (Equation 3) and P-(Equation 5) the same variable? If yes, standardize, use the same symbology.
216-218: “And the acceptable level of a CR value was usually from 10-4~10-6 [7]. When values of CR were > 10-6, the cancer risk was considered unacceptable…”
Comment: Reference values should be revised. 10-4 is greater than 10-6. How is it possible that values between 10-6 and 10-4 are acceptable and values greater than 10-6 are unacceptable?
300-301: “Analysis of the water samples indicated that the maximum concentration of Zn and Cd were at Harbin section”
Comment: Which sample point is associated with the Harbin section? S4 and S5? It has to be explicit in the text.
313: Figure 2
Comment: Improve the quality (resolution) of the figure.
329-331: “According to the pollution assessment standard proposed by Nemerow [29], the average of the single factor pollution index (Pi) of the six heavy metals in sediments and riparian soil has the same order: Cd > Zn > Cr > Ni > Cu > Pb”
Comment: I suggest that you insert a table (may be in Supplementary Information) with all the Pi values.
393-394: “Albasel and Cottenie had found that the contents of Pb, and Zn rapidly decrease with increasing distance from roads [16,49].”
Comment: Albasel and Cottenie (1985) are related only with reference 49. Rewrite the sentence.
Author Response
Authors' response to reviewers' comments
Manuscript ID: ijerph-728418
Type of manuscript: Article
Title: Concentrations, possible sources and health risk of heavy metals in multi-media environment of the Songhua River, China
Comments and Suggestions for Authors
There are some comments which have to be considered before publication.
110: “(3) to estimate daily heavy metal absorption (CDI)…”
Comment: Better to remove CDI from the sentence or use the correct meaning. Chronic Daily Absorption was used for CDI (line 202); however, CDI is most commonly used for Chronic Daily Intake. Does the CDI mean Chronic Daily Absorption or Chronic Daily Intake?
Response:
Thanks for the valuable comment. The CDI represents “Chronic Daily Intake”, while we have revised this expression in the introduction and following section (line 109 and 209).
130: Sample Collection and Processing
Comment: Give us more detail about the procedure and equipment used to collect the sediment.
Response:
Surface sediment (0-10 cm) was collected using a grab sampler from the 14 sampling sites using the same approach as for the water samples. For sediments, five sub-samples were combined into a single representative sample per site. Each sample was stored in a pre-washed glass container with a Teflon cap. (line 140-143).
187-189: Equations (3) and (5)
Comment: Are Pi(ave) (Equation 3) and P-(Equation 5) the same variable? If yes, standardize, use the same symbology.
Response:
It has been corrected (line 196 and 198).
216-218: “And the acceptable level of a CR value was usually from 10-4~10-6 [7]. When values of CR were > 10-6, the cancer risk was considered unacceptable…”
Comment: Reference values should be revised. 10-4 is greater than 10-6. How is it possible that values between 10-6 and 10-4 are acceptable and values greater than 10-6 are unacceptable?
Response:
We have revised this expression in the manuscript. The acceptable level of a CR value was usually between 10-6 and 10-4. When values of CR exceed 10-4 this indicates a high cancer risk to humans, while values of CR between 10-6 and 10-4 indicate a lower but elevated cancer risk. (line 223-226). And the ‘3.3 Human Health Risk Assessment’ and ‘4 Conclusion’ have been also revised (line 367-368 and line 445-446).
300-301: “Analysis of the water samples indicated that the maximum concentration of Zn and Cd were at Harbin section”
Comment: Which sample point is associated with the Harbin section? S4 and S5? It has to be explicit in the text.
Response:
Sampling sites S4 and S5 are associated with the Harbin section. We have added these in the manuscript (line 303). And detail information about the locations of sampling site was also presented in Supplementary Information (Table S1).
313: Figure 2
Comment: Improve the quality (resolution) of the figure.
Response:
The resolution of the Figure 2 has been improved.
329-331: “According to the pollution assessment standard proposed by Nemerow [29], the average of the single factor pollution index (Pi) of the six heavy metals in sediments and riparian soil has the same order: Cd > Zn > Cr > Ni > Cu > Pb”
Comment: I suggest that you insert a table (may be in Supplementary Information) with all the Pi values.
Response:
This information on Pi values has been added in Supplementary Information (Table S5).
393-394: “Albasel and Cottenie had found that the contents of Pb, and Zn rapidly decrease with increasing distance from roads [16,49].”
Comment: Albasel and Cottenie (1985) are related only with reference 49. Rewrite the sentence.
Response:
Corrected.

Reviewer 2 Report
Line 131 and following. How were the samples collected? On foot? From a boat? Were sediments collected with some kind of grab sampler?
Nothing is said about the time of the year in which the sampling was performed and this in my opinion is an important issue.
Line 140. If the sub-samples were combined (at least sediment and soil, nothing is said about water), where do the authors get the variability to make statistical comparisons?
Lines 244, 245. Can the authors explain how this comparison was made? Where do they get the necessary variability for the test in the background concentrations? The same for lines 259, 260 and 261.
Lines 272, 273, 274. The same issue as before, how the authors compare their data with those of Yangtze River? Where do they get the variability?
Line 332. Does it make sense to say that zinc was moderately polluted? Can an element be polluted?
Line 415. “Pb… it is hard to be degraded”. This phrase is not correct. Lead, like any non-radioactive element, does not degrade.
Line 425. It would be advisable that the points represented in Figure 4 had a projection towards the base to more easily appreciate the three-dimensionality.
Author Response
Authors' response to reviewers' comments
Manuscript ID: ijerph-728418
Type of manuscript: Article
Title: Concentrations, possible sources and health risk of heavy metals in multi-media environment of the Songhua River, China
Comments and Suggestions for Authors
Comment:
Line 131 and following. How were the samples collected? On foot? From a boat? Were sediments collected with some kind of grab sampler?
Nothing is said about the time of the year in which the sampling was performed and this in my opinion is an important issue.
Response:
The samples were collected in July 2015. We have added the detailed information on the manuscript “Surface sediment (0-10 cm) was collected using a grab sampler from the 14 sampling sites using the same approach as for the water samples. For sediments, the five sub-samples were combined into a single representative sample per site. Each sample was stored in a pre-washed glass container with a Teflon cap. At the same time, riparian soil samples (0-20cm) were also collected within 30 m at water sampling sites using a stainless steel scoop that had been prewashed with deionized water and stored in a pre-washed glass container”. Please see page 7-8, line 140-146.
Comment:
Line 140. If the sub-samples were combined (at least sediment and soil, nothing is said about water), where do the authors get the variability to make statistical comparisons?
Response:
Total 14 water samples were collected from the mainstream of the Songhua River. At each water sampling site, we also collected corresponding sediment and riparian soil samples. The detailed information on sample collection and procedure was presented in the line 136-146.
Comment:
Lines 244, 245. Can the authors explain how this comparison was made? Where do they get the necessary variability for the test in the background concentrations? The same for lines 259, 260 and 261.
Response:
The statistics method can generally prove the significance between two data further. In this study, the one-sample T-test was used to compare the concentrations of heavy metals in individual environmental media with background value for each heavy metal, and the comparison was necessary for understanding the pollution status of heavy metals in the mainstream of the Songhua River, including water, sediment, and riparian soil.
Comment:
Lines 272, 273, 274. The same issue as before, how the authors compare their data with those of Yangtze River? Where do they get the variability?
Response:
We used the statistics method to compare the monitored data of individual heavy metal from our study to the published data from Yangtze River or other Rivers in the world, which can better reflect the pollution degree in our study areas. Therefore, the one-sample T-test was used to compare one set of data of heavy metals from the Songhua River with those of Yangtze River.
Comment:
Line 332. Does it make sense to say that zinc was moderately polluted? Can an element be polluted?
Response:
Thanks for your comment. This section has been corrected in line 333-335.
Comment:
Line 415. “Pb… it is hard to be degraded”. This phrase is not correct. Lead, like any non-radioactive element, does not degrade.
Response:
This sentence has been corrected in the manuscript (line 416-417) “Due to its environmental persistence, Pb is still an important indicator of transportation sources”.
Comment:
Line 425. It would be advisable that the points represented in Figure 4 had a projection towards the base to more easily appreciate the three-dimensionality.
Response:
Thanks for the valuable comment. The projection towards the base of each point represented in Fig. 4 has been add in the revised figure (line 427).

Reviewer 3 Report
The manuscript deals with the impact of heavy metals in multi-media environment of the Songhua River, China. The study could provide the basis information for a risk management strategy. Overall, the approach of the study in the manuscript is good and could be useful in the public domain, but the manuscript needs considerable revision to reach the public domain. Authors are suggested to address following comments in order to make the manuscript suitable for publication.
* Abstract should be rewritten by detailing the aim and concept of the paper. More quantitative information should be provided in Abstract. Also add the benefits of the study findings and recommendations at the end of the abstract.
* Provide significant words which are more relevant to the work in logical sequence as ‘keywords’.
* Introduction is very general and need to be elaborative to explore the actual philosophy to design the experiment. The introduction is insufficient to provide the state of the art in the topic. Hypothesis should be given. How this work is different from the available data?
The originality and novelty of the paper need to be further clarified. What progress against the most recent state-of-the-art similar studies was made in this study?
* The last paragraph or closing lines of the introduction section always highlight the novelty aspects of the study with the clear aim of the study and the importance/significance of the study findings
*Line no 115; prove coordinates of the study area.
*The manuscript does not provide interesting and technically sound discussion; it would be better to use more recent references in discussion.
* Authors are suggested to add discussion by explaining trends in the obtained results along with the possible mechanisms behind the trends.
*Under section, discussion, it is recommended to discuss and explain what the appropriate policies should be based on the findings of this study. Also, the results should be further elaborated to show how they could be used for real applications.
* Authors are suggested to draw major inferences/primary conclusions first quoting the data/results obtained followed by the secondary conclusions/ recommendations reached through the critical analysis/ investigation of the study. Based on the outcome of the study, the author(s) may recommend the extension of the present study as the future scope of research.
Author Response
Authors' response to reviewers' comments
Manuscript ID: ijerph-728418
Type of manuscript: Article
Title: Concentrations, possible sources and health risk of heavy metals in multi-media environment of the Songhua River, China
Comments and Suggestions for Authors
The manuscript deals with the impact of heavy metals in multi-media environment of the Songhua River, China. The study could provide the basis information for a risk management strategy. Overall, the approach of the study in the manuscript is good and could be useful in the public domain, but the manuscript needs considerable revision to reach the public domain. Authors are suggested to address following comments in order to make the manuscript suitable for publication.
Comment:
* Abstract should be by detailing the aim and concept of the paper. More quantitative information should be provided in Abstract. Also add the benefits of the study findings and recommendations at the end of the abstract.
Response:
Thanks for the comments. We have revised the abstract section and provided necessary quantitative information and supplemented findings of this study for improving the scientific significance.
Comment:
* Provide significant words which are more relevant to the work in logical sequence as ‘keywords’.
Response:
We have revised the keywords for better reflecting the study objectives and content.
Comment:
* Introduction is very general and need to be elaborative to explore the actual philosophy to design the experiment. The introduction is insufficient to provide the state of the art in the topic. Hypothesis should be given. How this work is different from the available data?
Response:
Thanks for the valuable comments. We have revised this section, and provided the details about the concept of our study (details in the revised manuscript page 4-5). “For few current studies sought understand pollution from a more a wholistic multi-media perspective which were prefer looking at the pollutants in the context of single source-pathway-receptor relationships. Additionally, in non-occupational settings, it has been demonstrated that exposure is not dominated by a single pathway or exposure media. Therefore, a comprehensive human health risk assessment that integrates all important exposure pathways within the multi-media environment is needed to improve confidence in risk estimates. The present study was designed to provide the scientific basis for developing a multi-media, multi-pathway exposure risk model, as well as investigate potential risks in a more wholistic way and explore the characteristics and sources of heavy metal pollution in the multiple environmental media from the Songhua River catchment”.
Comment:
The originality and novelty of the paper need to be further clarified. What progress against the most recent state-of-the-art similar studies was made in this study?
Response:
Thanks for your comments. Introduction section has been revised and added the originality and novelty of our study. Compared to the previous study, the objectives of our study were mainly to understand pollution from a more wholistic multi-media perspective, and provided a more comprehensive understanding of the interaction between human activities and the riverine environment. The detailed information on revision presented in the introduction, results and discussion, and conclusions sections.
Comment:
* The last paragraph or closing lines of the introduction section always highlight the novelty aspects of the study with the clear aim of the study and the importance/significance of the study findings
Response:
The novelty of the study has been addressed in the last paragraph of the introduction section. “The results of this study will provide the scientific basis for developing a multi-media, multi-pathway exposure risk model. For the specific case of heavy metals, it provides a scientific rationale for identifying their spatial distribution and sources within the multi-media environment, as well as the important exposure pathways and associated risks”. (line 112-116)
Comment:
*Line no 115; prove coordinates of the study area.
Response:
The coordinates of the study area were added in line 122.
Comment:
*The manuscript does not provide interesting and technically sound discussion; it would be better to use more recent references in discussion.
Response:
Thanks for your comments. We have carefully revised the manuscript from all aspects including novelty, significance, technology, and language through the whole context. And we think that the revised version can meet the request on science, novelty and preciseness from journal and reader. The detailed revisions have been marked with red in the revised manuscript.
Comment:
*Authors are suggested to add discussion by explaining trends in the obtained results along with the possible mechanisms behind the trends.
Response:
We have added the necessary discussion to explain the possible influence on riverine environment or potential ecological or health risk, according to the obtained results. Such as “The HPI value was especially acute at location S6 which was influenced by Harbin City (upstream) as well as by diffuse sources from adjacent agricultural land” (Line 329-330), “In addition, the Pi of Cd in riparian soil is 3.54 times greater than the Pi(ave) at site S10, which suggests the presence of a specific point source or sources. Overall, values of HPI of water indicated that Cd and Pb were the main contributors to the derived level of pollution; while values of PN indicated that Cd and Zn were the most important pollution factors both in sediments and riparian soil” (Line 345-350), “Although the HPI and PN indicated that levels of Pb pollution in the catchment were low, its potential health risk to humans was indicated as being more significant highlighting the non-linear relationship between environmental levels and magnitude of exposure” (Line 360-363), and “The source apportionment confirmed that agriculture and industry were the main sources of heavy metals in the Songhua River basin. Thus, optimization and control of agricultural practices such as adopting precision agriculture approaches to chemical usage would aid in pollution mitigation. Due to the dominance of agricultural land use in the catchment, the potential for pollution reduction is considerable. At the same time, there are opportunities to adapt industrial processes and waste management approaches to improve their environmental sustainability. This is especially pertinent to large urban areas such as Harbin. It is also imperative to improve regulation of industrial units outside the main urban areas where clandestine discharges are likely more prevalent to improve the overall pollution level of locations such as S9” (Line 429-438).
Comment:
*Under section, discussion, it is recommended to discuss and explain what the appropriate policies should be based on the findings of this study. Also, the results should be further elaborated to show how they could be used for real applications.
Response:
We have added the appropriate policies based on the findings of this study. “optimization and control of agricultural practices such as adopting precision agriculture approaches to chemical usage would aid in pollution mitigation. Due to the dominance of agricultural land use in the catchment, the potential for pollution reduction is considerable. At the same time, there are opportunities to adapt industrial processes and waste management approaches to improve their environmental sustainability. This is especially pertinent to large urban areas such as Harbin. It is also imperative to improve regulation of industrial units outside the main urban areas where clandestine discharges are likely more prevalent to improve the overall pollution level of locations such as S9” (Line 429-438).
Comment:
* Authors are suggested to draw major inferences/primary conclusions first quoting the data/results obtained followed by the secondary conclusions/ recommendations reached through the critical analysis/ investigation of the study. Based on the outcome of the study, the author(s) may recommend the extension of the present study as the future scope of research.
Response:
We have revised the manuscript and addressed the significance of the study and the extension in the future research in the future according to the comments. “The results of the Songhua River water, sediments and soils confirmed that environmental pollution should be considered in a wholistic manner given the spatial variability of sources, interactions between pathways of exposure, as well as the non-linearity of the exposure term. Overall, the possibility of non-carcinogenic risk in the Songhua River was found to be very low. However, the cancer risk associated with consumption of the river water was slightly elevated above the cancer risk threshold. This cancer risk was mainly attributable to the presence of Cd in the water, and further work is required to understand the efficacy of the current water treatment regime for the removal or dilution of Cd and other potential pollutants associated with human health. The multi-media analysis indicated a significant accumulation of metals, particularly Cd and Zn, over time. Industrial emissions are likely to be the primary source of the observed heavy metal enrichment. Zn and Pb are also likely to have been derived from agricultural activities and transportation. Agricultural sources of Cd also cannot be ignored. Optimization and control of agricultural management with a focus on precision agriculture approaches could be one way to reduce pollution discharges. By adopting a more wholistic multi-media, multi-exposure approach to risk assessment, we have obtained a more comprehensive understanding of the interaction between local human activities and the riverine environment. This improved understanding helps aid mitigation responses as well as highlighting important knowledge gaps for future investigation” (Line 440-457).

Reviewer 4 Report
1) The authors have reported the concentration trend, possible sources and associated health risks of different heavy metals in water, sediment and soil of Songhua River, China. In case of drinking water supply, the heavy metals detected were under acceptable range however, concentrations in soils and sediments were exceeding values (might be leading to carcinogenic effect). However, I don't find any significant contribution in current research as it deals with reporting concentration values even at different multimedia environment and identifying potential sources of heavy metals penetration/disposal. Therefore, the authors must look into the novelty of current study.
2) One of the major observation is the concentration in different environmental media doesn't followed same order. Can authors explain why? Moreover, why authors reported in abstract that the concentration follow same order.
Zn > Cr > Cu > Pb > Ni > Cd (in water)
Zn > Cr > Ni > Pb > Cu > Cr (in sediments)
Zn > Cr > Ni > Pb > Cu > Cd (in soil)
3) Moreover, the authors must look at the format of journal carefully since line spacing is not as per format.
4) In the Figure 3, why s6 and s10 sampling site showed much higher HPI index in water and soil, respectively when compared with other media.
In general, the manuscript can be accepted for publication after addressing the above mentioned points.
Author Response
Authors' response to reviewers' comments
Manuscript ID: ijerph-728418
Type of manuscript: Article
Title: Concentrations, possible sources and health risk of heavy metals in multi-media environment of the Songhua River, China
Comments and Suggestions for Authors
Comment:
1) The authors have reported the concentration trend, possible sources and associated health risks of different heavy metals in water, sediment and soil of Songhua River, China. In case of drinking water supply, the heavy metals detected were under acceptable range however, concentrations in soils and sediments were exceeding values (might be leading to carcinogenic effect). However, I don't find any significant contribution in current research as it deals with reporting concentration values even at different multimedia environment and identifying potential sources of heavy metals penetration/disposal. Therefore, the authors must look into the novelty of current study.
Response:
Thanks for the valuable comment. As for many current studies about analyzing the regional pollution trends were always focused on single environmental media, which may omit the important information for sources analysis or health risk assessment. Our study was devoted to understand pollution from a more wholistic multi-media perspective, and provided a more comprehensive understanding of the interaction between local human activities and the riverine environment. We have rewritten the ‘Abstract’, ‘Introduction’ and ‘Conclusion’ section, and supplemented necessary discussion in the revised manuscript.
Comment:
2) One of the major observations is the concentration in different environmental media doesn't followed same order. Can authors explain why? Moreover, why authors reported in abstract that the concentration follow same order.
Zn > Cr > Cu > Pb > Ni > Cd (in water)
Zn > Cr > Ni > Pb > Cu > Cr (in sediments)
Zn > Cr > Ni > Pb > Cu > Cd (in soil)
Response:
The average concentration of heavy metals in sediments were followed the same trend observed in soil, but there were some differences compared to that of water. Thus, we changed “a similar trend” into “a different trend” in abstract (line 33).
And the heavy metals in sediments and soil tend to be the result of long-term accumulation, while those in surface water more closely reflect contemporary pollution within the catchment, thus explaining the relative differences seen between water and sediments. And the average concentrations of heavy metals in water were found significantly higher than the background concentrations of the Songhua River, indicating that higher emission intensities of wastewater and agricultural runoff.
Comment:
3) Moreover, the authors must look at the format of journal carefully since line spacing is not as per format.
Response:
The whole manuscript and Supplementary Information were adjusted to meet the format of double-spaced.
Comment:
4) In the Figure 3, why s6 and s10 sampling site showed much higher HPI index in water and soil, respectively when compared with other media.
Response
Thanks for the comment. The HPI value was especially acute at location S6 which was influenced by Harbin City (upstream) as well as by diffuse sources from adjacent agricultural land (Line 329-330).
The highest PN value occurred in S10 in riparian soil, for the Pi of Cd in riparian soil is 3.54 times greater than the Pi(ave) at site S10, which enhanced the whole pollution levels of S10, suggesting the presence of a specific point source or sources (line 345-347).
In general, the manuscript can be accepted for publication after addressing the above metioned points.

Round 2
Reviewer 2 Report
The manuscript can be accepted in its present form.
Reviewer 3 Report
The authors have addressed all the comments with full justification. Hence, the paper may be accepted in its current form